# Epithelial Mesenchymal Transition and Immune Response in Metaplastic Breast Carcinoma

**DOI:** 10.3390/ijms22147398

**Published:** 2021-07-09

**Authors:** Silvia González-Martínez, Belén Pérez-Mies, David Pizarro, Tamara Caniego-Casas, Javier Cortés, José Palacios

**Affiliations:** 1Clinical Researcher, Hospital Ramón y Cajal, 28034 Madrid, Spain; silviagonzalezmartinezbio@gmail.com; 2Department of Pathology, Hospital Ramón y Cajal, 28034 Madrid, Spain; bperezm@salud.madrid.org; 3Institute Ramón y Cajal for Health Research (IRYCIS), 28034 Madrid, Spain; david.pizarro@salud.madrid.org (D.P.); tamara880723@hotmail.com (T.C.-C.); 4CIBER-ONC, Instituto de Salud Carlos III, 28029 Madrid, Spain; 5Faculty of Medicine, University of Alcalá de Henares, Alcalá de Henares, 28801 Madrid, Spain; 6Faculty of Biomedical and Health Sciences, Department of Medicine, Universidad Europea de Madrid, 28670 Madrid, Spain; 7International Breast Cancer Center (IBCC), Quironsalud Group, 08017 Barcelona, Spain; 8Medica Scientia Innovation Research, 08007 Barcelona, Spain; 9Medica Scientia Innovation Research, Ridgewood, NJ 07450, USA; 10Vall d’Hebron Institute of Oncology, 08035 Barcelona, Spain

**Keywords:** MBC, metaplastic breast carcinoma, EMT, epithelial-mesenchymal transition immune system

## Abstract

Metaplastic breast carcinoma (MBC) is a heterogeneous group of infrequent triple negative (TN) invasive carcinomas with poor prognosis. MBCs have a different clinical behavior from other types of triple negative breast cancer (TNBC), being more resistant to standard chemotherapy. MBCs are an example of tumors with activation of epithelial–mesenchymal transition (EMT). The mechanisms involved in EMT could be responsible for the increase in the infiltrative and metastatic capacity of MBCs and resistance to treatments. In addition, a relationship between EMT and the immune response has been seen in these tumors. In this sense, MBC differ from other TN tumors showing a lower number of tumor-infiltrating lymphocytes (TILS) and a higher percentage of tumor cells expressing programmed death-ligand 1 (PD-L1). A better understanding of the relationship between the immune system and EMT could provide new therapeutic approaches in MBC.

## 1. Introduction

There is a close relationship between the immune system and the development of cancer. The immune response aims to destroy cancer cells and create a long-term immune memory. Both the innate immune system and adaptive immunity intervene in this anti-tumor immune response. In the innate system, Natural Killer (NK) cells are the main effectors, as a rapid response independent of antigen that is nonspecific and where there is no intervention of memory cells. However, the adaptive immune system generates a later and specific response, dependent on antigen, and memory cells are involved, specifically CD8 + cytotoxic T lymphocytes (CTLs) [1].

In the last few years, a role of the epithelial mesenchymal transition (EMT) process in the regulation of tumor immune response has been recognized. The relationship between EMT and the immune system is crossed. On the one hand cells of the innate and adaptive immune systems, such as macrophages, myeloid-derived suppressor cells (MDSCs), NK, and Tregs can induce EMT in tumor cells by secreting cytokines, inflammatory factors, and chemokines [2,3,4]. On the other hand, the mesenchymal-like tumor cells act on the immune system in two ways, by immunosuppression and evasion [5,6,7].

EMT involves the molecular and phenotypic changes that characterize the conversion of immobile cancer epithelial cells into mobile mesenchymal cells, with the characteristics of stem cells [8,9]. Whereas EMT in cancer can appear as a transient process in the invasive front of some tumors as well as during migration that favors the metastatic process, permanent EMT occurs in a small group of malignant human tumors in different organs. Although they receive different names depending on their localization, all are characterized by a biphenotypic appearance including both malignant epithelial (carcinomatous) and mesenchymal (sarcomatous) components. In some cases, the malignant epithelial component can be minimal and limited to the non-invasive (in situ) part of the tumor. The two more characteristics examples of this group of tumors are uterine carcinosarcoma (Mixed Mullerian Malignant Tumor) [10] and metaplastic breast (MBC) carcinomas [11].

MBC is a predominantly triple negative (TN) and morphologically heterogeneous group of invasive carcinomas that display heterologous differentiation of the neoplastic epithelium towards squamous cells and/or mesenchymal-type elements such as spindle, chondroid, and osseous cells [12]. Five histological subtypes are recognized in the new WHO classification of breast tumors: squamous cell carcinoma (SqCC), spindle cell carcinomas (SpCC), MBC with heterologous mesenchymal differentiation (MBCHMD), low-grade adenosquamous carcinoma (LGASC), and fibromatosis-like MBC (FLMBC) [12]. In addition, mixed MBC (MMBC) is composed of more than one subtype. MBCs can be subdivided into low-grade and high-grade (HG-MBC) according to their behavior. HG-MBC includes SqCC, SpCC, and MBCHMD. In this review, we focus on HG-MBC. In addition, although pure SqCC does not represent a typical example of EMT, we have included this subtype in this review since most tumors with squamous cell differentiation also carry some degree of spindle cell or heterologous differentiation.

MBC has specific clinical and pathological features. We reviewed the clinical and pathological features of 32 MBC series [13,14,15,16,17,18,19,20,21,22,23,24,25,26,27,28,29,30,31,32,33,34,35,36,37,38,39,40,41,42,43,44], with a total of 11,066 tumors. Around 80% of MBC were TN. In contrast, hormone receptor positivity and HER2 positivity was reported in 0–13% and 0–10% of the series respectively. Tumors were usually large, with a mean size of 3.3 cm. In these series, MBC was diagnosed in patients ranging from 47 to 67 years old, with an average age of around 58 years. In one series, important age differences among histological subtypes were observed, MBC occurring with chondroid differentiation at a mean age of 71 years, whereas SpCC and SqCC presented at mean ages of 56 and 48 years respectively [27]. Most tumors presented in stage II and histological grade 3. About 32% of the tumors had lymph node metastases and 11% visceral metastasis at diagnosis. Furthermore, lymphovascular invasion was observed in nearly 11% of the tumors.

Referring to the prognosis, this was analyzed in 15 of the 32 reviewed series [20,21,22,23,24,25,29,33,35,36,37,38,39,41,44]. Most of them compared survival of MBC with survival of other types of breast cancer (BC). In general, the prognosis was significantly worse in patients with MBC than in patients with other types of BC including non-metaplastic TNBC. The 5-years and 3-years OS in MBCs ranged from 50% to 89% and 66% to 76.6%, with an average of 67.7% and 72.3% respectively.

Conventional therapeutic approaches in MBCs include surgery, chemotherapy, and radiotherapy. However, the clinical response to systemic therapies is limited due to partial resistance of MBC to conventional chemotherapy. For this reason, despite there being some new approaches such as mTOR inhibitors, check-point inhibitors, or PARP inhibitors, new targeted therapies based on specific molecular features or immunomodulation are required.

## 2. Epithelial-Mesenchymal Transition in Metaplastic Breast Cancer

### 2.1. Cadherins and Cadherin Switching

Epithelial cells exhibit adherens junctions, apico-basal polarity, and limited migratory potential. During EMT, epithelial cells progressively miss their cell identity and morphology and increasingly acquire mesenchymal characteristics [45,46]. One of the first events in the EMT process is cadherin switching, by which epithelial cells lose the expression of *E*-cadherin and express other mesenchymal cadherins.

Proteins of the cadherin family are crucial mediators of cell–cell adhesion and modulation of cadherin expression is closely related to EMT [47]. Cadherins possess diverse structures and functions and any alteration on its structure or function can lead to breast malignancy [48].

The type I cadherins are the best characterized subgroup of cadherins and include epithelial (*E*)-, neural (*N*)-and placental (*P*)-cadherin (*CDH1*, *CDH2*, and *CDH3*, respectively) [49]. Their correct function and stability require the interaction with members of the catenin protein family, α-, β- and γ-catenin, through their cytoplasmic tails and p120-catenin at their transmembrane region [50,51].

*E*-cadherin, the prototypical member of the type-1 classical cadherins, is a key mediator of cell–cell adhesions in epithelial tissues, while reduced or loss of *E*-cadherin expression by inactivating mutations, promoter methylation or transcription regulation, can induce invasive and metastatic behavior in many epithelial tumors [52]. Although loss of E-cadherin is a hallmark of EMT, in BC this alteration typically occurs in lobular cancer, both in situ and invasive, a tumor type that does not have other typical characteristics of EMT, indicating that *E*-cadherin loss is necessary but not sufficient to develop a complete EMT program. In invasive ductal BC, the level of *E*-cadherin expression is related with the grade of differentiation [53].

In normal human tissues, the expression of *P*-cadherin partially overlaps with the expression of *E*-cadherin. *P*-cadherin expression is typical of regenerative epithelial layers, mostly basal, whereas *CDH1* is mainly expressed in suprabasal layers and simple epithelia [54,55]. In fact, several studies have clarified that the expression of *P*-cadherin is crucial for the maintenance of normal mammary epithelial architecture [56,57]. In cancer, the function of P-cadherin is clearly context dependent. In melanoma, *P*-cadherin behaves like a tumor suppressor gene, in which there is a progressive loss of normal *E*-cadherin and *P*-cadherin expression from melanocytes, followed by an increase in *N*-cadherin expression in melanoma [58,59]. However, in other tumors, especially in BC, P-cadherin has been associated with an increase in cell invasion and tumor aggressiveness and is a factor that indicates a worse prognosis. Furthermore, many studies have shown that aberrant expression of P-cadherin is characteristic of TNBC [60,61].

*N*-cadherin, contrary to previous cadherins, normally functions in non-epithelial tissue and is considered to be a mesenchymal cadherin. In epithelial cells, aberrant *N*-cadherin expression contributes to weak adherents junction and promotes motility, invasion, and metastasis. In these cells, *N*-cadherin interacts with the FGF-receptor (*FGFR*), activating the *MAPK/ERK* cascade. *N*-cadherin overexpression in BC is correlated with invasiveness as a result of *N*-cadherin-mediated interactions between cancer and stromal cells and is also characteristic of TNBC [62].

Cadherin-11 is a type II cadherin. It mediates cell-to-cell homophilic interactions and is mainly expressed on mesenchymal cells, it is essential for tissue migration and organization during embryogenesis [63,64]. In highly malignant forms of breast and prostate cancer, cadherin-11 expression increases migration and invasion of tumor cells [65]. The expression of cadherin-11 may be well correlated with the invasive phenotype in BC cells and may serve as a molecular marker for the more aggressive, invasive subset of tumors [66]. Moreover, cadherin 11 expression has been correlated to the basal B group of BC [67].

Lien et al. [68] and Sarrió et al. [67] first reported EMT in MBC and demonstrated by immunohistochemistry loss of *E*-cadherin expression in the metaplastic mesenchymal component (spindle cells and chondroid and osseous differentiation), a finding subsequently reported in other studies [27,33]. In contrast E-cadherin was retained in the epithelial component. In addition, since most MBC are TNBC, *P*-cadherin expression was also frequent in the epithelial component. Both, E- and P-cadherin are expressed in the squamous component of MBC, indicating the preservation of the epithelial phenotype.

In contrast to *E*- and *P*-cadherin, N-cadherin and cadherin-11, which were not expressed in the epithelial component of MBC, were expressed in the mesenchymal component in a high percentage of tumors. This process of cadherin switching was accompanied by altered expression of catenins. As a result of the loss of *E*-cadherin, α-, β- and γ-catenin tend to be also lost, whereas accumulation of p120-catenin occurs in the cytoplasm. Cytosolic p120-catenin controls tumor growth and survival through the regulation of Rho GTPase and integrin survival [69]. In some tumors, β-catenin that is not bound to membranous E-cadherin can be transferred to the nucleus where it can activate Wnt target genes, such as c-*MYC* or *CCND1* [70].

### 2.2. Epithelial and Mesenchymal Markers

Several immunohistochemical and transcriptomic studies have shown that in MBCs there is down-regulation of epithelial markers [27,67,71,72,73] and up-regulation of mesenchymal markers [27,67,68,72,73,74,75], as a manifestation of EMT. See Figure 1.

Downregulated epithelial markers in MBC belong to different families, such as keratins (181), claudins, occludins, and others. McQuerry et al. [73] demonstrated that *CD24*, keratinocyte-related genes such as *CALML5* and *KRT81*, and the late cornified envelope genes, *LCE1F*, *LCE3D*, and *LCE3E*, were down-regulated in MBC but were expressed in most TNBC samples. Piscuoglio et al. [27] showed that MBCs with spindle cell differentiation show distinct transcriptomic profiles compared to squamous and chondroid MBCs. Mainly, spindle cell subtype carried down-regulated genes related to EMT and decreased expression of genes such as *CDH1* and *EPCAM* in this subgroup. They also revealed decreased expression of tight junction related genes in this subgroup, including *CLDN3*, *CLDN8*, *CGN*, *MYH11*, and *MYH14*.

During the transition to a mesenchymal phenotype, the cytokeratin network anchored to the desmosomes of the epithelial cells is destroyed and the cytoskeleton is reorganized due to the increased expression of mesenchymal proteins such as vimentin [76]. The transition to a mesenchymal state facilitates cell motility and the formation of new membrane protrusions. Finally, it results in extra cellular matrix degradation, cell migration, and invasive behavior. Moreover, trans-differentiation to specific mesenchymal tissues, such as cartilage or bone may occur. Accordingly, MCBs overexpressed mesenchymal genes functionally related to cytoskeletal remodeling, extracellular matrix synthesis and remodeling, cell adhesion, motility, and migration, as well as genes associated with skeletal development and/or chondro-ossification *(SPARC, MMPs, VIM*, etc) [27,67,68,73]. In addition, MBC have markedly elevated *CD44/CD24* and *CD29/CD24* ratios and *ALDH- 1* expression, which are characteristic of stemness [72,74,75]. See Figure 2.

### 2.3. Epithelial-Mesenchymal Transition—Transcriptional Factors

Many transcriptional repressors of E-cadherin have been identified. The main ones include SNAIL family zinc finger transcription factors (*SNAIL1* and *SNAIL2*), the zinc finger E-box binding homeobox proteins (*ZEB1* and *ZEB2)*, and the TWIST family basic helix–loop–helix transcription factors (*TWIST1* and *TWIST2*) [77]. *SNAIL1/SNAIL2* and *ZEB1/ZEB2* downregulate the expression of a number of target genes by binding to E-box DNA sequences through their carboxy-terminal zinc-finger domains [78,79,80,81,82]. *TWIST1/TWIST2* bind as dimers and recognize cis-regulatory elements, called hexanucleotide sequence E-boxes, to function as transcriptional factors (TFs) [83].

The expression of EMT-TFs can be overlapping, and they can form networks, yet their functions are usually distinct. They can induce both common and specific genetic programs, suggesting a differential role of the factors in EMT [84]. Thus, for example Snail1 and Zeb1 are mutually required for EMT induction while continuous Snail1 and Snail2 expression, but not Zeb1, is needed for maintenance of the mesenchymal phenotype in some in vitro models [85].

All EMT-TFs play an important role in the modulation of cadherin expression. All of them downregulate *E*-cadherin and upregulate *N*-cadherin [47]. In addition, these TFs also coincide in the downregulation or upregulation of other proteins. For example, *SNAIL1/SNAIL2, ZEB1/ZEB2* downregulate plakophilin but upregulate *MMP*s. *SNAIL1/SNAIL2* downregulate claudins, occluding, desmoplakin, cytokeratins, and plakophilin and upregulate fibronectin and several collagens. Additionally, *ZEB1/ZEB2* downregulate *ZO1* and *TWIST1* upregulates *α5integrin*. These TFs are also regulated by common signaling pathways. For example, *SNAIL1/SNAIL2* and *ZEB1/ZEB2* are regulated by TGFβ-smad3 and Wnt/β-catenin pathways. In addition, *SNAIL1/SNAIL2* are also modulated by *NOTCH, PI3K-AKT, NF-kβ, EGF* and *FGF* pathways [46,78,79,86,87,88,89,90,91,92,93,94,95,96,97,98,99,100].

Several series analyzed the expression of different EMT-TFs in MBC [33,72,101,102,103,104,105]. Transcriptomic studies reported that the TFs that are most frequently overexpressed in the sarcomatous component of MBC are *SNAI1, SNAI2, ZEB1*, and *TWIST1* [72,74,101,102,103,106,107]. In addition, by immunohistochemistry, Zhou et al. [106] demonstrated nuclear accumulation of Slug and Twist in 78.6% and 93% of the SpCC and 100% of the matrix-producing carcinomas. In contrast, Oon et al. [101] found *TWIST* overexpression in 57.1% of the MBCs tumors, however, in their study, SqCC was the most frequent (42.9%) metaplastic component. In the study of Nassar et al. [107], 100% of MBC, including all histological subtypes, expressed *SNAIL*. Finally, *ZEB1* overexpression was detected in 41% of MBCs in the study by Zhang et al. [74]. In this series, all the *ZEB1* positive tumors exhibited mesenchymal differentiation towards spindle (86%), chondroid (83%), or osseous (100%) elements. The glandular and squamous areas were negative for *ZEB1* expression in all cases.

### 2.4. miRNAs

EMT is regulated by different miRNAs through their interaction with EMT-TFs and direct modulation of gene expression. Several in vitro and in vivo studies have suggested a core miRNA signature associated with EMT [85]. The miR-200 family (miR-200f), which members include miR-200a, miR-200b, miR-200c, miR-141, and miR-429, is the principal component of this signature. The miR-200 members share many of their targets, due to the high sequence homology between them in their seed region. Their overexpression leads to increased *E*-cadherin expression, the maintenance of the epithelial phenotype, and the inhibition of EMT [108]. The miR-200f plays a major role in regulating epithelial plasticity, mainly through its involvement in double-negative feedback loops with the EMT-TFs *ZEB1*, *ZEB2*, *SNAI1*, and *SNAI2*, ultimately influencing the expression *E*-cadherin and other genes [102,109,110].

There are few studies analyzing miRNA expression in MBC. Gregory et al. [109] and Díaz-Martín et al. [85] reported reduced expression of miR-200f members in MBC when compared to other histological types of BC. In addition, Sánchez-Cid et al. [111] demonstrated a decreased in miR-200f members expression in the sarcomatous component of MBC when compared to the epithelial component.

miR-200f members can be regulated not only by EMT-TFs but also at the transcriptional level by promoter hypermethylation. Specifically in MBC, Castilla et al. [102] reported, miR-141 promoter hypermethylation as a mechanism of gene silencing. In vitro studies have demonstrated the miR-200f methylation is, at least partially mediated by the acquisition of EMT features in these tumors.

Although not specifically studied in MBC, other miRNAs have been linked with EMT and the development of stem-cell properties in TNBC. Thus, miR-10b, miR-21, miR-29, miR-9, miR-221/222, and miR-373 tend to be overexpressed, whereas miR-145, miR-203, and miR-205 [112,113] tend to be hypo-expressed in these tumors.

### 2.5. Genetic Alterations

Several oncogenic pathways cooperate in the initiation and progression of EMT via cytoskeleton reorganization and activation of *E*-Cadherin repressors. The main pathways are *TGF-β*, canonical *WNT*, and *NOTCH*. Other signal transduction pathways implicated in EMT induction are, among others, Hippo*-YAP/TAZ*, *TP53*, and *PI3K/AKT* signaling pathways.

In a review of 14 series of MBCs [13,15,16,17,18,26,27,28,30,31,32,42,44,114], with a total of 539 tumors, we observed that the genes most frequently mutated were *TP53* and several genes of the *PI3K* pathway, such as *PIK3CA*, and of the *WNT* pathway, such as *APC* (see Figure 3). The most frequent mutations in the *TP53* and *PIK3CA* genes are represented in Figure 4 and Figure 5. Regarding copy number variations (CNVs), the genes most affected were *CDKN2A/B* with loss in 18.7% and 19% of the cases respectively, and *CNDD3* that was amplified in 15% of the cases (see Figure 3). Interestingly, genetic alterations in *TGF-β* and Hippo pathways were infrequent, suggesting indirect modulation of these pathways through genetic alterations in others. For example, in the study of Díaz-Martín et al. [85], practically 100% of MBCs with histological evidence of EMT toward spindle cell or heterologous sarcomatous differentiation showed overexpression of the Hippo effector *TAZ*, in spite of the lack of genetic alterations in this pathway.

### 2.6. Therapy Response in MBC

As mentioned above, conventional therapeutic approaches in MBC include surgery, chemotherapy, and radiation therapy. Surgery continues to be the standard therapy, in most case series, while the most used surgical approach is mastectomy, which is due to the large size of the tumor at the time of presentation. However, if the tumor is small, conservative surgery and radiotherapy can be performed without affecting disease-free survival or overall survival, since T-stage seems to be the most significant prognostic factor [115,116,117,118]. The current clinical evidence on the effect of radiotherapy in improving outcomes in patients with MBC is limited, the studies that exist are retrospective, but RT should continue to be an integral component of the multidisciplinary management of patients with MBC undergoing both breast-conserving surgery and mastectomy [117]. At the time of presentation, MBC are usually in an advanced stage, so chemotherapy, both neoadjuvant and adjuvant, is especially important, although the results are usually poor. There is no established regimen, some of the administered regimens are based on doxorubicin and ifosfamide as neoadjuvant treatment or based on anthracycline and taxane as adjuvant treatment [115,117,119].

The poor prognosis and poor response to neoadjuvant therapy of MBC suggested chemoresistance associated to EMT. Several in vitro studies have demonstrated that resistance to anthracyclines and taxanes—the two types of drugs usually used in TNBC—is, at least in part, mediated by EMT. Indeed, both *SNAIL* and *TWIST1* can induce resistance to doxorubicin via upregulation of P-glycoprotein. Similarly, *SNAIL*-mediated upregulation of *PARP1* in human MDA-MB-231 BC cells also contributes to doxorubicin resistance. *ZEB1*, by stimulating an AKT/GSK3B/β -catenin pathway, also facilitated resistance to doxorubicin [120]. Similarly, acquired resistance to taxanes in human breast tumors and established cell lines is associated with the appearance of EMT phenotypes secondary to activation of a *TWIST1**/**AKT2* signaling axis or to *NOTCH*-dependent upregulation of Snail or Slug [120].

Few targeted therapies have been used in patients with MBC. The high frequency of alterations in the PI3K/AKT/mTOR pathway makes MBC a good candidate for treatment with mTOR inhibitors. Actually, treatment with mTOR inhibitors has been more successful in MBC than in other types of TNBC [121,122]. The combination therapy of everolimus, an mTOR inhibitor, plus cisplatin, which interferes with DNA function, has recently been shown to be effective in the neoadjuvant setting in three patients with MBC, who achieved RCB-I at the surgery [123]. MBCs in a patient with *BRCA2* germline mutation developed a complete pathological response after treatment with a *PARP* inhibitor [124].

Different inhibitors directed at EMT have been tested in BC, in high-throughput screening, preclinical trials, and clinical trials. These treatments could hypothetically impact on both tumor growth and chemo resistance. However, in spite of very promising results of anti-TGF-β therapy in cancer models, outcomes observed in many cancer clinical trials failed to recapitulate the preclinical data [125]. Similarly, the performance of anti-NOTCH therapies, such as gamma secretase inhibitors, in clinical trials generally has not reflected their encouraging results in preclinical studies [126].

## 3. Immune Response and Immune Therapy in MBC

In vitro studies have demonstrated differences in the immune response in tumors according to EMT status. Thus, tumors derived from mammary carcinomas with an epithelial phenotype express high levels of MHC-I, low levels of programmed cell death ligand-1 (PD-L1) and contained within their stroma CD8 + T cells and M1 macrophages (anti-tumor). In contrast, tumors arising from carcinoma cell lines with a mesenchymal phenotype and exhibiting EMT markers expressed low levels of MHC-I, high levels of PD-L1, and contained within their stromal regulatory T cells, M2 macrophages (tumor), and exhausted CD8 + T cells (see Figure 6). Furthermore, mesenchymal carcinoma cells within a tumor retained the ability to protect their epithelial counterparts from immune attack. Finally, epithelial tumors were more susceptible to elimination by immunotherapy than the corresponding mesenchymal tumors [127].

Dongre et al. [128] demonstrated that mesenchymal carcinoma cells exert immunosuppressive effects by modulating the tumor microenvironment (TME). These cells secreted immunosuppressive factors, such as *CD73*, *CSF1*, or *SPP1*, which reduced the number and function of CD8 + T-cells [128]. Hsieh et al. [129] demonstrated that the EMT-TF Snail directly activates the transcription of *MIR21* to produce miR-21-abundant tumor-derived exosomes. The miR-21-containing exosomes were engulfed by CD14+ human monocytes, suppressing the expression of M1 markers and increasing that of M2 markers [129]. In addition, in vivo studies demonstrated that in TNBC, cancers cells secreted pro-inflammatory cytokines that promoted conversion of monocytes to macrophage cells which, in turn, stimulated EMT, proliferation, chemoresistance, and motility in cancer cells [130].

In addition to inducing immunosuppression, EMT can also promote immune evasion by increasing the expression of checkpoints. For example, Noman et al. [131] showed that *ZEB1* overexpression in BC cells not only promoted EMT, but also, through modulation of miR-200, was able to induce PD-L1 expression in tumor cells. Overexpression of *SNAI1* or *ZEB1* in epithelial MCF7 cells activated EMT and up-regulated *CD47. CD47* expressing cells were less efficiently phagocytized by macrophages [132].

Several studies have evaluated the expression of PD-L1 in MBCs. Although there were methodological and result differences among the series, all of them demonstrated that a high percentage of MBC (30 to 70%) expressed PD-L1 in tumor cells. In addition, most studies also observed higher expression of PD-L1 [13,30,133,134,135,136,137] in MBC than in other types of TNBCs. [30,133,135]. In contrast, no significant difference in the expression of PD-L1 in stromal cells was found among different types of tumors [133,135,137] (see Table 1). Additionally, no differences in PD-L1 expression in tumor cells were observed among different subtypes of MBC [135,138]. Lien et al. [138] reported an increase of PD-L1 expression in immune cells of MBC with squamous differentiation.

The amount of stromal tumor infiltrating lymphocytes is a prognostic and predictive marker in BC. Abundant tumor-infiltrating lymphocytes (TILs) predict good prognosis and chemotherapy response in both HER2-positive and TNBCs. TILs have been evaluated in different series of MBCs. Kalaw et al. [135] reported that the MBCs showed a significant reduction in the proportion of TILS compared to TNBC, and Morgan et al. [133] demonstrated reduction of CD8 expression on immune cells in MBC samples in comparison to TNBC samples. On the other hand, MBCs were enriched for TILs expressing FOXP3 [135].

It has been suggested that there are differences in the immune microenvironments among MBCs subtypes. Thus, in the study by Joneja et al. [30], although total TILs were not enumerated, it was noted that TILs vary greatly within the MBC cohort by histologic examination. Chao et al. [134] found sqCC MBC exhibiting most TILs of all the MBC subtypes. Similarly, Lien et al. [138] reported that sqCC exhibited the highest positivity rate of TILs, while matrix-producing carcinoma had the lowest. Kalaw et al. [135] did not observe significant association between TILs and morphological subtypes of MBCs but observed that PD-L1 positive TILs were enriched in squamous cell carcinomas. Since squamous cell carcinoma is composed of epithelial tumor cells, whereas mesenchymal tumor cells predominate in MBCs, these observations suggested a role of EMT factors in controlling the abundance of TILs.

There is controversy regarding the association between TILS and longer disease-free survival in MBCs. Chao et al. [134] and Chien et al. [138] found a relationship between a greater number of TILS and longer disease-free survival, but Kalaw et al. [135] found that TILs content did not significantly impact BC-specific survival in their MBC cohort. The latter authors observed that an increased amount of FOXP3 positive intra-tumoral TILs was associated with an adverse prognostic outcome [135].

Referring to tumor mutational burden (TMB), Tray et al. [17] observed that the majority of MBC had a low TMB in a study with a total of 192 cases of MBCs, with a median of 2.7 mutations/Mb (range 0–39.6). Only 2% of tumors showed a high TMB (score 15 or greater). TILs were more frequently observed in high versus low TMB MBC, although the difference was not significant.

On summarizing all these data, it was evident that the immune response in MBCs differed from other TNBCs. The high proportion of PD-L1 expression make this tumor type a good candidate for immunotherapy with checkpoint inhibitors. There are case reports demonstrating complete response of MBCs to pembrolizumab and durvalumab [139]. In addition, there are ongoing clinical trials in which MBCs can be treated with check-point inhibitors. For example, an MBC cohort (Arm 36) is included in the DART study (NCT02834013) that evaluates dual anti-CTLA-4 (ipilimumab) and anti-PD-1 (nivolumab) blockade [140]. Additionally, the therapy with atezolizumab for TNBC (including MBC) containing 1% PD-L1 positive immune cells in the tumor biopsy has been approved, based on the IMpassion130 clinical trial (NCT02425891) [13].

## 4. Conclusions

Considering the poor response of MBC to conventional chemotherapy, new targeted therapies are needed. Whereas therapies directed to the *PI3K/AKT/mTOR* have obtained promising results, new approaches such as EMT or immune response modulation, should be attempted in MBC.

## Figures and Tables

**Figure 1 ijms-22-07398-f001:**
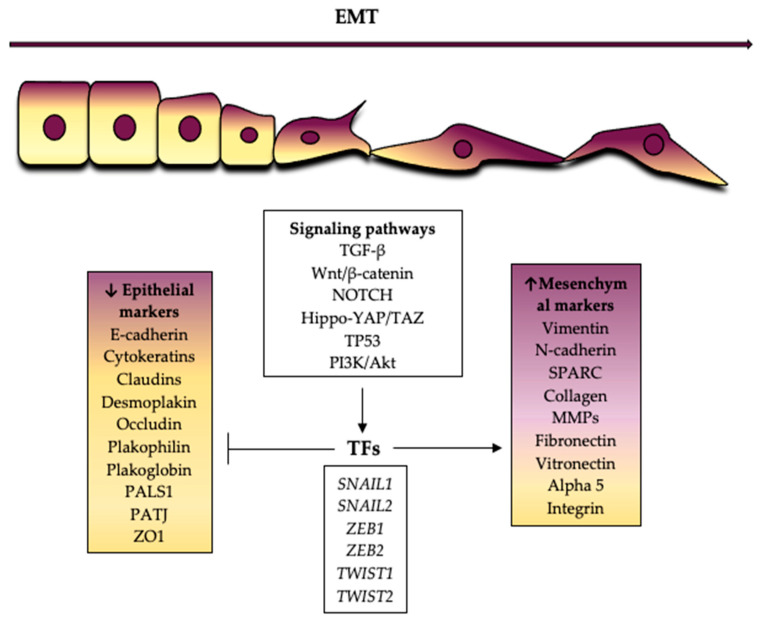
Epithelial and mesenchymal markers whose expression is decreased or increased during epithelial mesenchymal transition.

**Figure 2 ijms-22-07398-f002:**
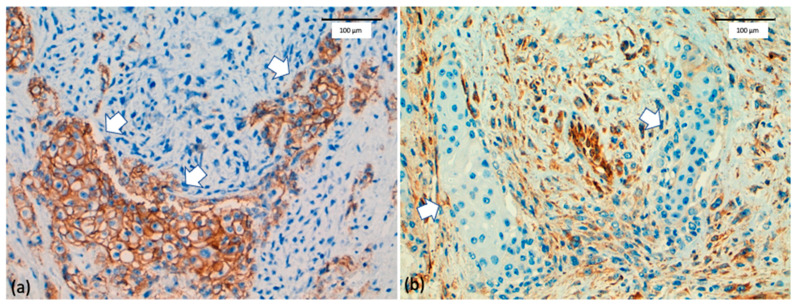
(**a**) E-cadherin expression is preserved in the epithelial component (arrows) and lost in the mesenchymal component of matrix-producing metaplastic breast cancer (scale bar 100 µm). (**b**) Intense SPARC expression in the mesenchymal component of matrix-producing metaplastic breast cancer. The epithelial component (arrows) is negative for SPARC (scale bar 100 µm).

**Figure 3 ijms-22-07398-f003:**
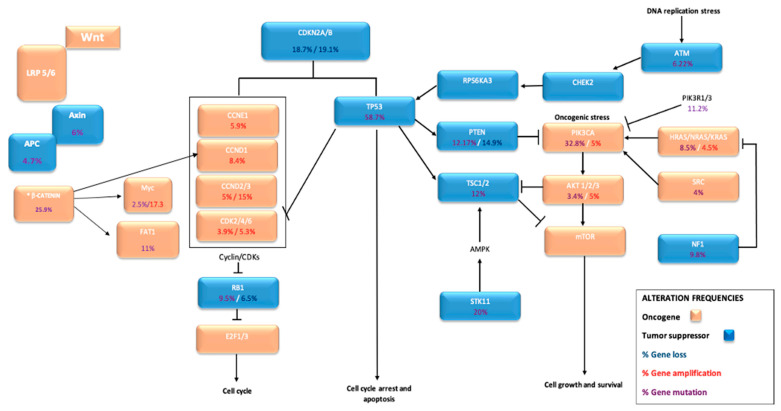
The most frequently altered genes in metaplastic breast cancer. * Mutation frequency reported by Hayes et al. [42] and not found by other authors. These data have been obtained from the review of the reference [11].

**Figure 4 ijms-22-07398-f004:**
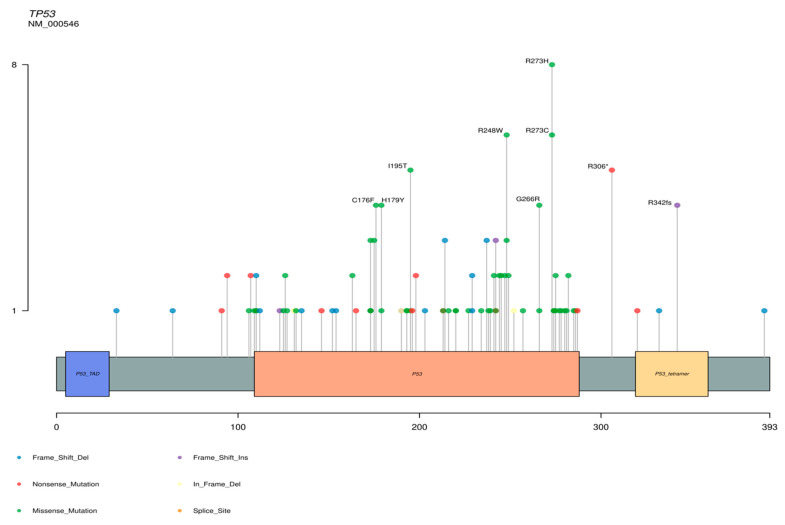
Representation of the distribution of mutations in the *TP53* gene. (*) Nonsense mutation. (fs) FrameShift mutation. These data were obtained from the review of reference [11].

**Figure 5 ijms-22-07398-f005:**
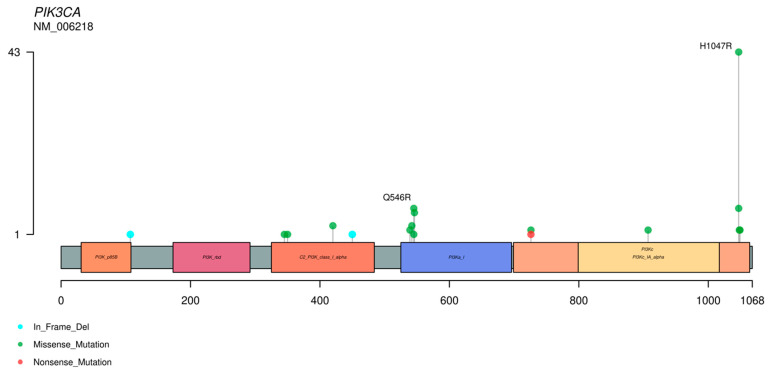
Representation of the distribution of mutations in the *PIK3CA* gene. These data were obtained from the review of reference [11].

**Figure 6 ijms-22-07398-f006:**
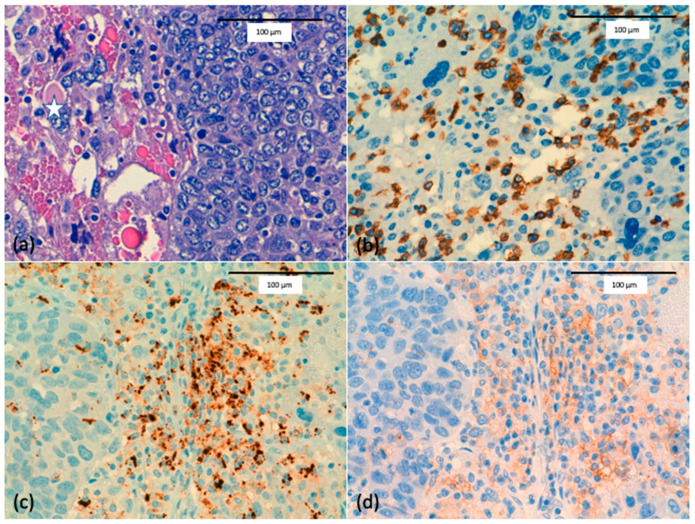
(**a**) Metaplastic breast cancer with a mesenchymal pleomorphic component (start) (scale bar 100 µm). (**b**) CD8 immunostain highlights the intratumoral immune cells in pleomorphic metaplastic breast cancer (scale bar 100 µm). (**c**) Programmed cell death ligand-1 (PD-L1) (SP 142 clone) immunostain showing dark-brown punctate and linear staining mostly expressed in intratumoral immune cells and in sarcomatoid tumoral cells (scale bar 100 µm). (**d**) PD-L1 (22C3 clone) immunostain showing moderate membranous staining in intratumoral immune cells and in sarcomatoid tumoral cells (scale bar 100 µm).

**Table 1 ijms-22-07398-t001:** Programmed cell death ligand-1 (PD-L1) expression in tumor cells and tumor microenvironment of metaplastic breast cancer, in some cases comparing data with other BCs.

		MBC	TNBC	Others	Significant Difference
Kalaw et al. [135]	(Antibody clone E1L3N)	*n* = 145	*n* = 79		
PD-L1% in tumor ≥5%	73%	~18%		Yes
PD-L1% in immune cells ≥5%	63%	63%		No
Morgan et al. [133]	(Antibody clone SP263)	*n* = 27	*n* = 119		
PD-L1% in tumor ≥ 1%	29.6%	10.1%		Yes
PD-L1% in immune cells ≥ 1%	59.3%	73.1%		No
Stephen et al. [137]	(Antibody clone E1L3N)	*n* = 12	*n* = 18		
PD-L1% in immune cells ≥5%	41.5%	38%		No
Joneja et al. [30]	(Antibody clone SP142)	*n* = 72		*n* = 218	
PD-L1% in tumor ≥5%	46%		6–9%	Yes
PD-L1% in immune cells ≥5%	43%			
Lien et al. [138]	(Antibody clone SP142)	*n* = 82			
PD-L1% in tumor ≥1%	17%			
PD-L1% in immune cells ≥1%	47.5%			
Chao et al. [134]	(Antibody clone SP142)	*n* = 60			
PD-L1% in tumor ≥ 1%	50%			
PD-L1% in immune cells ≥ 1%	60%			
Vranic et al. [13]	(Antibody clone SP142)	*n* = 23			
PD-L1% in tumor ≥ 1%	30.4%			
PD-L1% in immune cells ≥ 1%	8.7%			
Dill et al. [136]	(Antibody clone SP142)	*n* = 5			
PD-L1% in tumor ≥ 1%	40%			
PD-L1% in immune cells ≥5%	80%

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
