# Peer review of "Epithelial Mesenchymal Transition and Immune Response in Metaplastic Breast Carcinoma"

_ijms, 2021, doi:10.3390/ijms22147398_

Round 1

Reviewer 1 Report

The authors of the manuscript ‘’Epithelial Mesenchymal Transition and Immune Response in Metaplastic Breast Carcinoma’’, tried to review the status of a difficult topic in breast cancer, focusing on the EMT and Immune response aspects of these tumors.

The overall approach is excellent, but there are always minor suggestions that I will try to point out to the authors, in order to improve this review further and make it easier for the readers to follow.

FIGURES 2 and 5:

On the top right corner of each IHC picture there is a scale bar, which is blurry. Please consider changing this and mention the exact scale bar size in the figure legend.

Moreover, for consistency, please try to use the same magnification for all IHC pics. Last, but very important, the PDL1 staining shows a fragmented and occasionally not specific staining. Since PDL1 is expressed not only by certain immune cell types, like macrophages, but also by tumor cells and stromal cells, please consider presenting a more representative IHC picture for this marker.

FIGURES 3 and 4:

The fond size of these two figures should be the same.

Please clarify why you chose to present the mutations/ gains only for WNT pathway and not for TGF-beta, or NOTCH. The word ‘’gains’’ refers to the term ‘’gain of function’’ or overexpression/amplification?

Conceptually, taking into consideration the therapeutic aspect of MBC, that you present, I would suggest to consider also including the mutational profile of the PI3K/ AKT /c-MYC pathway, as a figure.

Finally, one last suggestion would be, to add a paragraph regarding the therapeutic options that we currently have, (surgery- chemotherapy-radiation) and then expand on the novel approaches like the small molecule inhibitors and immunomodulation.

Author Response

Dear reviewer,

Thank you very much for your review of our work. We greatly appreciate your comments that will improve the quality of our MS.

Response to Comments and Suggestions for Authors

1.- The authors of the manuscript ‘’Epithelial Mesenchymal Transition and Immune Response in Metaplastic Breast Carcinoma’’, tried to review the status of a difficult topic in breast cancer, focusing on the EMT and Immune response aspects of these tumors.

The overall approach is excellent, but there are always minor suggestions that I will try to point out to the authors, in order to improve this review further and make it easier for the readers to follow.

We appreciate the generous comments from the reviewer.

2.- FIGURES 2 and 5:

On the top right corner of each IHC picture there is a scale bar, which is blurry. Please consider changing this and mention the exact scale bar size in the figure legend.

Moreover, for consistency, please try to use the same magnification for all IHC pics. Last, but very important, the PDL1 staining shows a fragmented and occasionally not specific staining. Since PDL1 is expressed not only by certain immune cell types, like macrophages, but also by tumor cells and stromal cells, please consider presenting a more representative IHC picture for this marker.

We have modified these images, marking the scale bar in better resolution. This is specified in the legend. We have also taken the photographs at the same magnification, 200X in figure 2 and 400X in figure 5 (now figure 6). In this last figure, we have added other images that complement the previous ones.

3.- FIGURES 3 and 4:

The fond size of these two figures should be the same.

We have decided to remove figure 4 and include the WNT pathway to the left of figure 3.

4.- Please clarify why you chose to present the mutations/ gains only for WNT pathway and not for TGF-beta, or NOTCH.

We have not represented the TGF-beta, or NOTCH pathways due to the low frequency of mutations in these pathways that seem to be affected more frequently at the transcriptional level.

5.- The word ‘’gains’’ refers to the term ‘’gain of function’’ or overexpression/amplification?

We use the term gain to refer to amplification. We have clarified this in section 2.4 Genetic alterations and in the legend of figure 3.

6.- Conceptually, taking into consideration the therapeutic aspect of MBC, that you present, I would suggest to consider also including the mutational profile of the PI3K/ AKT /c-MYC pathway, as a figure.

The PI3K/ AKT pathway is included in figure 3, we have also added c-MYC and we have made a diagram of PIK3CA (new figure 5) with the distribution of all mutations in this gene. We have included another diagram showing TP53 mutations (new figure 4).

7.- Finally, one last suggestion would be, to add a paragraph regarding the therapeutic options that we currently have, (surgery- chemotherapy-radiation) and then expand on the novel approaches like the small molecule inhibitors and immunomodulation.

We have added this paragraph in section 2.5 Therapy response in MBC.

Reviewer 2 Report

Very nicely written review. Metaplastic breast cancer is a subtype of breast cancer where prognosis remains poor and significant work needs to be done. 

Will recommend authors to add citation PMID: 33420229 in paragraph 2.5 as this manuscript illustrates response in metaplastic TNBC with mTOR inhibitor.

Author Response

Dear reviewer,

Thank you very much for your review of our work. We greatly appreciate your comments that will improve the quality of our MS.

Response to Comments and Suggestions for Authors

1.-Very nicely written review. Metaplastic breast cancer is a subtype of breast cancer where prognosis remains poor and significant work needs to be done. 

We appreciate the generous comments from the reviewer.

2.-Will recommend authors to add citation PMID: 33420229 in paragraph 2.5 as this manuscript illustrates response in metaplastic TNBC with mTOR inhibitor.

This paragraph has been added with the corresponding reference (123).

“The combination therapy of everolimus, a mTOR inhibitor, plus cisplatin, which interferes with DNA function, has recently been shown to be effective in the neoadjuvant setting in 3 patients with MBC, who achieved RCB-I at the surgery”.